# Uncovering temporal changes in Europe's population density patterns using a data fusion approach

Filipe Batista e Silva [1✉], Sérgio Freire [1], Marcello Schiavina [1], Konštantín Rosina[1,2], Mario Alberto Marín-Herrera[1], Lukasz Ziemba[1], Massimo Craglia[1], Eric Koomen [3] & Carlo Lavalle[1]

The knowledge of the spatial and temporal distribution of human population is vital for the study of cities, disaster risk management or planning of infrastructure. However, information on the distribution of population is often based on place-of-residence statistics from official sources, thus ignoring the changing population densities resulting from human mobility. Existing assessments of spatio-temporal population are limited in their detail and geographical coverage, and the promising mobile-phone records are hindered by issues concerning availability and consistency. Here, we present a multi-layered dasymetric approach that combines official statistics with geospatial data from emerging sources to produce and validate a European Union-wide dataset of population grids taking into account intraday and monthly population variations at 1 km$^2$ resolution. The results reproduce and systematically quantify known insights concerning the spatio-temporal population density structure of large European cities, whose daytime population we estimate to be, on average, 1.9 times higher than night time in city centers.

[1] European Commission, Joint Research Centre, Via E. Fermi 2749, 21027 Ispra, Italy. [2] Institute of Geography, Slovak Academy of Sciences, Štefánikova 49, 814 73 Bratislava, Slovakia. [3] School of Business and Economics, Vrije Universiteit Amsterdam, De Boelelaan 1105, 1081 HV Amsterdam, The Netherlands. ✉email: filipe.batista@ec.europa.eu

Knowledge of population distribution is crucial for spatial analysis and policy support in many domains. Yet, despite all the progress since the inception of the first modern censuses in the early nineteenth century and the emergence of digital cartography and Geographical Information Systems in the second half of the twentieth century, our knowledge of the spatio-temporal distribution of population remains remarkably incomplete.

The emergence of dasymetric mapping in the 1910s[1] and its rediscovery many decades later[2,3], thanks to increasing access to digital censuses, geospatial data and computing power[4] contributed substantially to improve the geographical representation of population distribution. Dasymetric mapping can be described as a smart areal interpolation method[5] that operates by disaggregating population counts usually available per administrative units or census zones to a finer set of zones using a covariate of population distribution available at higher spatial resolution. Examples of covariates typically include land use/land cover (LULC) features (e.g., built-up and roads) or properties (e.g., built-up density, soil imperviousness and nighttime lights) derived from remote sensing and other geospatial datasets[6–12], as well as user-generated content from social media[13,14].

Dasymetric mapping is often applied to generate population grids or tesselations of regular squared cells with estimates of population. Such grids help mitigate the distortions associated with the Modifiable Areal Unit Problem[15] to the extent that they increase the spatial resolution, are less arbitrary, and remove the original areal heterogeneity vis-à-vis the original population enumeration zones. Population grids have become essential inputs for the analyses of human–environment interfaces and to support a wide range of applications by national and local governments, non-governmental organizations, and companies, including regional and urban planning, disaster risk management, and geomarketing[16].

Currently, there are multiple gridded population products of varying spatial resolutions and characteristics, with global or continental coverages[4,17–19]. In countries lacking up-to-date and reliable official demographic data, small-area estimation of population becomes possible, thanks to the increasing availability of Earth Observation data and other emerging sources of (big) geospatial data, computational power, and new statistical techniques[20]. For example, in a recent application in Nigeria, population was estimated independently from national census data, employing a bottom-up modeling approach combining a detailed mapping of built-up areas and a survey of local population densities[21]. In Europe, reliable bottom-up population grids can be constructed by aggregating address-based population counts provided by National Statistical Institutes (as opposed to dasymetric, top-down grids), of which GEOSTAT 2011 is the most recent compilation. For a more thorough discussion of the differences between top-down and bottom-up approaches, please refer to the paper from Wardrop et al.[20], whereas for a recent review more centered on top-down methods for large-scale applications and their fitness for use, we recommend the paper by Leyk et al.[4].

The vast majority of population grids are based on place-of-residence population counts. These maps can be used as proxies for nighttime population distribution[22], assuming that most people stay in their declared places of residence at night for shelter and rest. However, population is a temporally dynamic variable, with major shifts in its distribution occurring in daily and seasonal cycles, resulting in rapidly changing densities. Consequently, studies requiring spatially detailed information on population distribution are constrained to a static and incomplete representation of this dynamic phenomenon. Shifting from place-of-residence to place-of-activity population grids allows us to produce spatially explicit representations of the present population for different temporal frames. Such information is helpful for applications where both the spatial and temporal dimensions of population density are important, such as transport planning, or assessment of human exposure to natural, environmental, epidemiological, and technological hazards[23–27].

Daytime population distribution varies greatly from that of nighttime. The location of population during the day is determined by the location of economic, social, leisure, and other facilities, which attract population from their residences, driving commuting flows, and other forms of trips[28,29]. Therefore, it is significantly more challenging to infer daytime population distribution than mapping nighttime population. First and foremost, there is no single, measurable statistical concept instrumental for daytime population as the number of residents is for nighttime population. When a person is not present in his place of residence, he or she is likely to engage in multiple activities during the day such as studying, working, shopping, and going for leisure. Fundamentally, the locations of the presence of people are a function of their activities. Although there are aggregate statistics describing the size of the groups of people engaged in various activities, these are rarely linked to the exact time and locations of the activity. Thus, daytime population needs to be inferred from multiple, indirect sources. This complexity helps explain the significant delay in the development of spatio-temporal population mapping vis-à-vis conventional resident population mapping.

The first studies trying to map daytime population distribution date back to the mid-twentieth century and used passenger counts in a number of cities in the United States[30,31]. The few recent case studies focused on relatively small study areas and employed different methodologies and input data. Approaches to map spatio-temporal population can be classified into two broad categories as follows: (a) the ones that combine different sources of statistical and geospatial data in a dasymetric manner, and (b) the ones that use direct geolocated measurements of population activity from mobile network operators, sensors, or social media.

Originally motivated by the needs of emergency response, the LandScan population grid was an early attempt at mapping spatio-temporal population at the global scale, at a resolution of 30 by 30 arc-seconds[32]. It combined census data with several geospatial datasets to map ambient population (an estimate of the average present population throughout the daily cycle). The concept of ambient population, however, is of limited value for applications requiring time-specific representations of population. Subsequent work achieved actual daytime population estimates for the whole of the United States[33,34] or individual cities[35]. In Europe, such estimates have been produced for relatively small areas or single countries[23,26,36]. Other research[37–41] focused on increasing the temporal resolution by combining statistics and micro data with detailed geospatial data yet, again, for specific regions or urban agglomerations. A recent study attempted to derive seasonal population variations in Greece from nighttime lights from Earth observation[42].

The recent emergence and increasing availability of unconventional, big geospatial data sources[43] creates new opportunities for assessing spatio-temporal population dynamics. Geotagged posts on Twitter have been used to assess cross-border mobility patterns[44]. Location history recorded by Android smartphones has been tested to assess human mobility at micro-level[45] and to characterize the structure of cities based on spatial mobility patterns[29].

One of the most promising data sources for spatio-temporal population originates from mobile network operators. Their data are generated by the interaction between mobile-phone terminals and geolocated mobile network towers, and can be used to analyze mobility patterns of mobile-phone users[28,46] and map

spatio-temporal population densities at potentially high temporal granularity[47–50] including for specific population groups such as tourists[51,52]. Although generally seen as a promising data source, its use for systematic and large-area applications remains unpractical for two main reasons. First, access to such data is still limited. Operators are reluctant to release their data because of privacy issues and lack of business models[53]. Even if some mobile-phone operators agree to participate in pilot studies[54], with the current legal frameworks it is not possible to guarantee data access from all operators across multiple countries simultaneously. A second important issue relates to data quality and consistency. Population estimates from either Call Detail Records or Signaling data from a particular mobile-phone operator only represent the population covered by that operator, whereas all operators miss people without mobile devices (e.g., very young or old population segments). On the other hand, double counting may occur when the same individual carries more than one mobile device. Other technical issues compromise the quality of the data, such as low antenna density in rural areas (leading to heterogeneous spatial resolution) and antenna switching in busy areas. Moreover, Call Detail Records are particularly sensitive to temporal uncertainty (non-continuous observations), as only certain types of events (e.g., calls) are captured. In sum, all these issues lead to an overall spatial and temporal uncertainty of population estimates[55]. Although various developments are underway to overcome these limitations (e.g., algorithms to correct data biases, switching from Call Detail Records to Signaling data, and trusted smart statistics frameworks to harmonize access to data from different operators[56]), substantial technical and organizational progress is still required for a more systematic use of these data[55]. For a more complete account and discussion of the state-of-the-art concerning estimation of spatio-temporal population, we recommend the recent review from Panczak et al.[27].

The primary objective of this work is to produce the first European Union (EU)-wide representation of spatio-temporal population distribution taking into account both the seasonal and intraday variations of population that is seamless and consistent across countries, and which can be accessed and used free of charge by researchers, policy officers, and practitioners in multiple fields. Due to the issues reviewed earlier, the use of Mobile Network Operator for this continental-scale exercise was unfeasible. Instead, we developed a multi-layered dasymetric approach that expands upon the generic dasymetric method by modeling the spatial distribution of different population groups separately and according to a selection of covariates obtained from emerging sources of geospatial data. The output of this approach consists of a set of 24 population grids, one daytime and one nighttime grid for each month of the year, at 1 km² resolution. We evaluated the quality these grids in four EU countries, where adequate independent data were available. We found a high level of agreement between estimated and reference population distribution, although generally higher for the nighttime period. The analysis of European cities with a population above 1 million ($n = 34$) revealed that, on average, daytime population densities are 1.9 times higher than nighttime densities in city centers and then decay exponentially with distance to city center.

## Results

**Multi-temporal population grids**. To produce the multi-temporal population grids, we downscaled monthly stocks of individual population groups at subnational level to grid-cell level using a population group-specific set of spatial covariates. The population groups included the number of residents, workers for different economic sectors, students, tourists, and non-working and non-studying population. The main stocks of population

groups were obtained from official statistics. The monthly variations in population stocks were derived from school calendars as well as from monthly inbound and outbound tourists from official statistics. To obtain the final monthly day- and nighttime population grids, we summed the respective monthly grids of specific population groups. For example, the daytime population grid for January corresponds to the sum of the previously generated grids for January of workers, students, tourists, and the non-working and non-studying population (refer to the "Methods" section for a more detailed description).

The resulting 24 population grids (or temporal frames) cover the 28 Member States of the EU (as of 2019) at a spatial resolution of 1 km², which was selected for its adequacy to support sub-regional and urban analyses. This set of multi-temporal population grids represent a typical working day of the month. The variation between workdays and weekend is not addressed. The nighttime frames represent an ideal situation assuming the whole population is at their place of residence or lodging to rest, whereas the daytime frames represent a situation whereby everybody is assumed to be at the location of their primary activity such as working or studying during core working hours. As such, in-between daily variations of population are not taken into account (e.g., commuting, pre- or after-work activities, etc.). The reference year for population data is 2011, to match with the latest round of the European censuses.

**Analysis of spatio-temporal patterns**. A three-dimensional rendering of the population density at nighttime and daytime for the city of Milan, Italy, and surrounding areas, based on the produced dataset, is displayed in Fig. 1. It reveals substantial differences in the distribution and concentration of population between the two periods. To further illustrate the results, Fig. 2 shows the absolute differences in population per 1 km² grid cells between day- and nighttime (yearly average), and between August and January (at night) for three selected areas in Europe. The top three maps provide a spatially explicit representation of daily variations in absolute population. For example, Paris, France, is characterized by a net gain in population in daytime in a rather large area corresponding to the city core, resulting from a large concentration of economic activities, surrounded by a belt of predominantly residential areas that lose population in daytime. Although much smaller, the city of Lisbon shows a similar pattern, whereas in Milan the areas with higher population densities in daytime appear more scattered. Differences between August and January also have very distinct spatial patterns. The historical core of Paris clearly gains population in August compared to January, whereas most of its surroundings display a net loss. Some positive hotspots are visible in areas such as the Charles De Gaulle airport and Disneyland. In the south of Portugal, population in August outweighs the population in January, both in the historical center of Lisbon and the in southernmost coastal areas of Algarve. Finally, in the North of Italy, all the Milan metropolitan area loses population in August, whereas gains are observed around the lakes Maggiore, Como, and, even more noticeably, Garda.

To further illustrate the insights that can be extracted from the produced dataset, we investigated some spatio-temporal characteristics of the largest urban agglomerations in Europe. To select them, we used the city/greater city extents defined by Eurostat[57]. This definition was designed to improve comparability of city statistics and applies a fixed set of criteria related to urban morphology, to consistently characterize city limits irrespective of national definitions. For each of the listed 800+ cities/greater cities, we summed the population in day- and nighttime based on yearly average grids and selected those whose day- or nighttime population is above 1 million people in 2011,

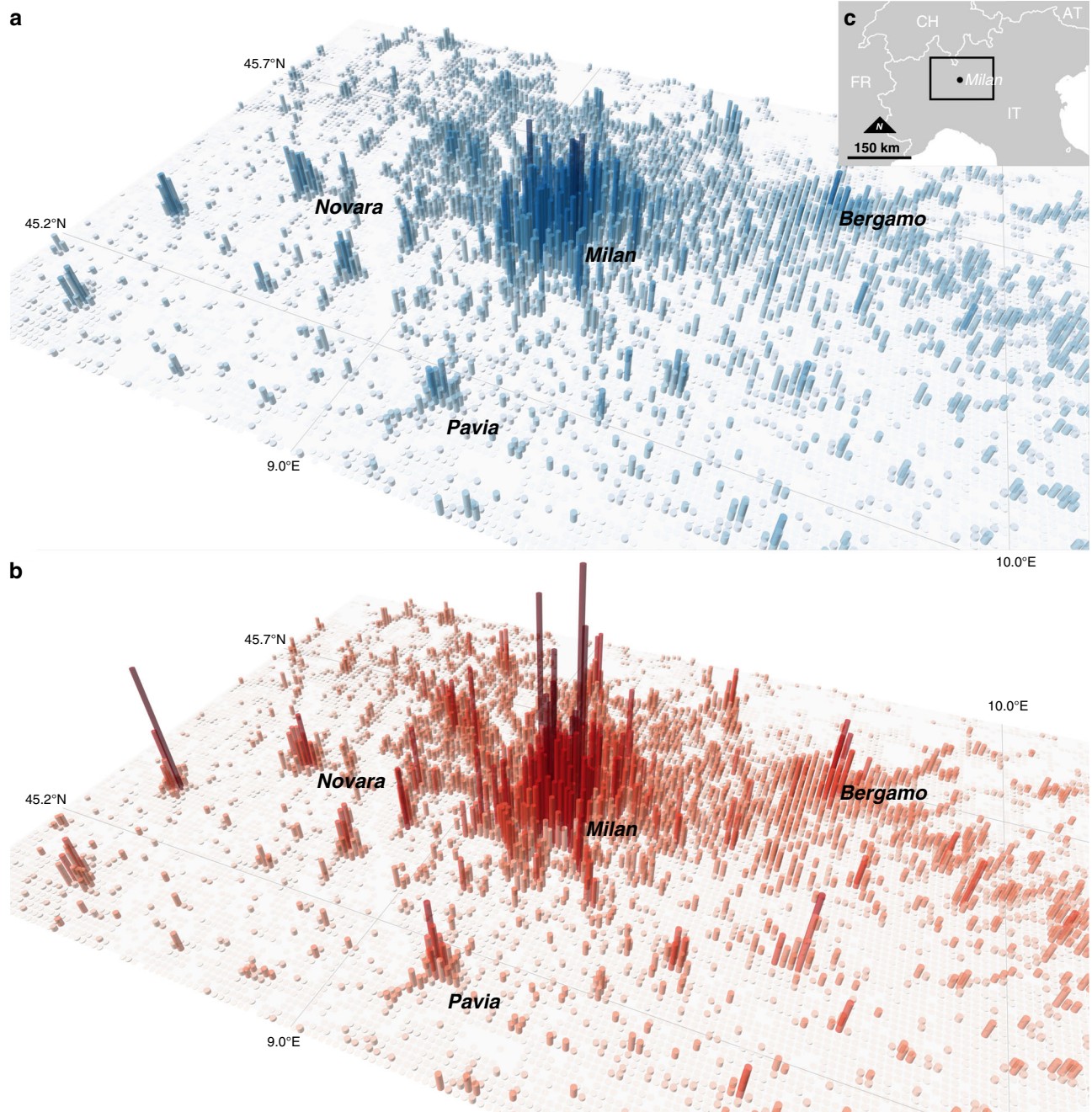

**Fig. 1 Bird's-eye view of population density in the area of Milan.** Population density based on the average of all months at nighttime (**a**) and daytime (**b**). These representations cover an approximate swath of 150 km (East–West) by 100 km (North–South), and each vertical bar corresponds to a 1 × 1 km grid cell. The height of bars is linear function of the estimated population. The highest bar at nighttime (**a**) records 23.3k persons and the highest bar at daytime (**b**) records 42.9k persons. **c** The geographical location and extent of the represented area. The figure can be reproduced using the publicly deposited multi-temporal population grids[75]. Copyright EuroGeographics for the administrative boundaries.

resulting in a sample size of $n = 34$. On average, daytime population outweighs nighttime population in Europe's largest urban agglomerations. The average day-to-nighttime ratio for the sampled cities is 1.097 ($\sigma = 0.098$). The highest ratios were observed in Budapest and Warsaw (1.31–1.32), followed by Brussels (1.24), whereas the three Spanish cities of Madrid, Barcelona, and Valencia, together with Athens and Stockholm, display surprisingly low ratios in the range 0.94–0.99. On average for the sampled cities, and based on yearly averages, the composition of the daytime population is 48.9% employees ($\sigma = 7.0\%$), 22.7% students ($\sigma = 3.0\%$), and 1.2% tourists ($\sigma = 0.7\%$).

The remainder 27.2% of the population correspond to the non-working and non-studying residents (Supplementary Table 1).

As the selected 34 cities have a relatively large size ($\bar{x} = 776.9\,\text{km}^2$, $\sigma = 464.4\,\text{km}^2$, see Supplementary Table 1), high within-city variations of diurnal and nocturnal population densities are expected. A common way to characterize urban densities is by creating profiles describing the decay of population densities as a function of distance to city centers[58]. With our multi-temporal population grids, it is possible to compare day- and nighttime population density profiles for European cities for the first time. In Fig. 3, we plot the average population density

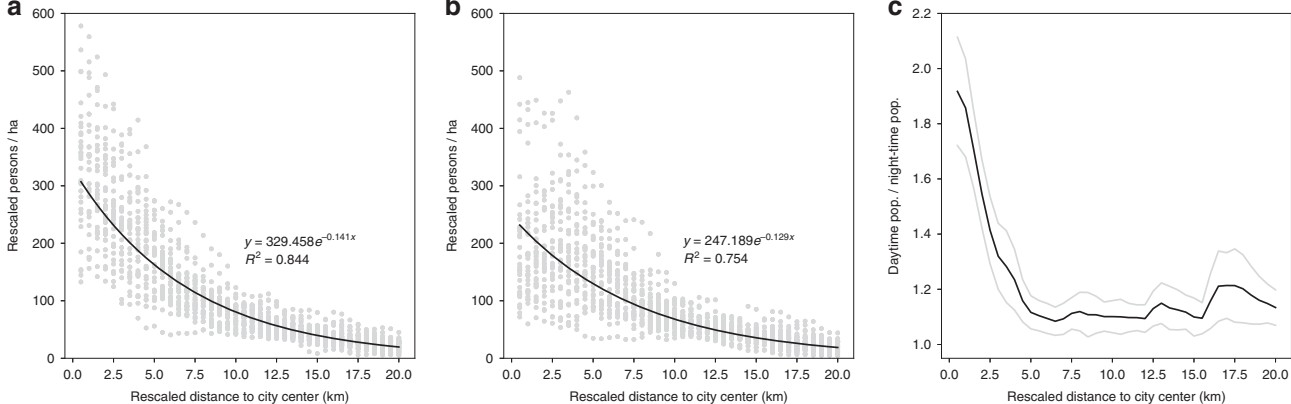

**Fig. 2 Spatio-temporal differences in population density in selected sites.** Absolute differences in population at 1 km$^2$ grid-cell level between day- and nighttime (average of all months) (**a**, **c**, **e**) and between August and January (at nighttime) (**b**, **d**, **f**) for south of Portugal (**a**, **b**), North of Italy (**c**, **d**) and Ile-de-France (**e**, **f**). The circles represent a radius of 20 km from the city centers of Lisbon, Milan, and Paris. The figure can be reproduced using the publicly deposited multi-temporal population grids[75]. Copyright EuroGeographics for the administrative boundaries.

**Fig. 3 Concentric population density profiles of the largest European Union cities.** Average population density for daytime (**a**) and nighttime (**b**), and average ratio between day- and nighttime populations (**c**) for the 34 cities in Europe with a population above 1 million in 2011. Population densities and distance to city center were rescaled as prescribed by Lemoy and Caruso[59] to make the concentric population density profiles comparable across cities of different population sizes. Gray dots in **a** and **b** represent individual city measurements of population density. Gray lines in **c** delimit the 95% confidence interval of the mean. In **a** and **b**, an exponential curve was fitted to the population density of all the 34 cities, resulting in a $R^2$ of 0.844 and 0.754, with $n = 1360$ (34 cities × 40 concentric ring measurements) and $p$-value < 0.0001, for daytime and nighttime, respectively. Source data are provided as a Source Data spreadsheet file.

gradients around city centers with a spacing of 500 m. These concentric rings typically take the town hall as center and consider only the land area to avoid distortions in coastal or waterfront cities. Population densities and distance to center were rescaled as prescribed by Lemoy and Caruso[59] to make the radial population density profiles comparable across cities of different population sizes, which, in our sample, range over one order of magnitude. The rescaling makes all cities comparable in dimension to the most populous city in the sample (i.e., Paris). Without rescaling, the curves cannot be compared across $x$ and $y$ axes, as more populous cities tend to be denser and extend over a larger geographical area to accommodate the extra population[59].

Consistent with recent literature[59–61], nighttime population densities are highest at or nearby the city center and then decay with increasing distance from the city center, fairly well described with a negative (inverse) power law function. Daytime population densities show a similar profile but densities at the city center are significantly higher than at nighttime and descend more abruptly. Although population density distribution shows great variation per city based on local conditions, its relation with the distance to city centers is remarkably stable within our ensemble of cities, as indicated by the high $R^2$ obtained of 0.844 and 0.754, with $n = 1360$ (34 cities × 40 concentric ring measurements) and $p$-value < 0.0001, for daytime and nighttime, respectively (Fig. 3a, b).

In addition, Fig. 3c plots the average profile of the ratio between day- and nighttime population, peaking at 1.9 in the city center and descending rapidly until a distance of 5 km from where it hovers just above 1. The spread around the mean widens after a distance of 15 km, owing to the diversity of local settlement geographies. Day- and nighttime density profiles for each sampled city can be found in Supplementary Figs. 2–6.

To get a first impression of typologies of spatio-temporal behavior in our sample of cities, we apply a clustering algorithm to find similarity between cities in the ratios of day- over nighttime population densities. We applied $k$-means that is a commonly used, straightforward partitional algorithm[62]. Our clustering relies on the rescaled distance from the city center of the first 30 rings (i.e., 15 km radius) on the $x$-axis and the ratio of densities on the $y$-axis. It is noteworthy that the clustering is applied on the aggregated one dimensional description of our cities and does not refer to its spatial distribution in geographical space for which other clustering approaches would be more appropriate (see, e.g., Sander et al.[63]). We chose four clusters based on the analysis of the within groups similarity for the result of the $k$-means run for a pre-specified number of clusters from 2 to 15. Figure 4 shows the resulting day-to-nighttime ratio profiles for the identified clusters. The cluster in Fig. 4b is the most distinct, as city centers appear to be predominantly residential and only towards the periphery daytime densities surpass nighttime densities. This is the smallest cluster, composed of the three Spanish cities in the sample (Madrid, Barcelona, and Valencia) plus Lyon, France. The other clusters all show a decreasing ratio from the center outwards. However, in the cluster in Fig. 4c the densities are much higher in the center than in cities belonging to the clusters in Fig. 4a, d. Conversely, the main difference between the clusters in Fig. 4a, d is that in the former the ratio picks up after 5 km, signaling major employment hubs or satellite cities close to the main city, whereas in the latter the ratio drops gently but steadily until a longer distance from the city center.

**Quality assessment**. To evaluate the reliability of the produced population grids, we calculated the allocation accuracy for areas where adequate reference data were available. We obtained census-based estimates of day- and nighttime population for the whole of Italy and Portugal per municipality. In addition, we obtained similar data specifically for three cities in Spain (Madrid, Barcelona and Valencia) to verify the reliability of the surprisingly low estimates of the day-to-nighttime ratios found for those cities. Finally, for Belgium, we estimated the day- and nighttime population based on data procured from a Mobile Network Operator. A validation of the monthly population variation was not possible to perform due to the lack of adequate data at sub-regional level. However, in our grids, regional seasonal curves of inbound tourists and school holidays are both based on official sources (see "Methods").

Table 1 contains the results of the cross-comparison. The results show an almost perfect agreement with the nighttime (i.e., residential) population records from the censuses of Italy, Portugal, and Spain. This result is not surprising, as our nighttime population distribution is identical to the census-based GEOSTAT grid (see "Methods" section). On the other hand, the daytime population grid obtained a very consistent allocation accuracy of nearly 93% in these countries. The comparison against mobile network operator-based data for Belgium reveals a lower degree of agreement. Two chief reasons explain this outcome: the substantially smaller size of spatial zones and the conceptual differences between our multi-temporal population grids and the human activity measurements obtained from this source. Notwithstanding, it is worthwhile noting a small spread between day- and nighttime accuracies, suggesting that the quality of the day- and nighttime grids is comparable.

For the three Spanish cities mentioned above, we calculated the day-to-nighttime population ratio based on the census data. The obtained ratios were 1.014 for Madrid, 1.006 for Barcelona, and 0.963 for Valencia, which compare to our estimates of 0.981, 0.954, and 0.948, respectively. Although our ratios appear lower than what census data suggest (likely due to an underestimation of daytime population density within the Spanish cities), the peculiarity of the Spanish cities is corroborated.

## Discussion

Considerable progress has been made in recent years in mapping place-of-residence population distribution at large scale[4]. Although of crucial importance for many applications[20], such grids are a limited representation of reality, roughly corresponding to population densities during nighttime. Assessments of population exposure to natural, health, and technological hazards, adequate planning of transport and social infrastructure in cities and regions, and the study of functional urban areas[64] require knowledge of the changing population distribution in temporal cycles resulting from human mobility.

Data from sources such as mobile network operators, sensors, or social media provide geolocated measurements of human activity at high spatio-temporal resolution, but have a number of limitations such as restrained data access and data inconsistencies. In this study, we developed an approach to model spatio-temporal population for large areas in a consistent manner, and that is not constrained by the latter limitations. The employed approach can be referred to as multi-layered dasymetric mapping and it combines official statistics on population groups (i.e., residents, workers, students, and tourists) with geospatial data from conventional (e.g., mapping agencies), as well as emerging data sources (e.g., voluntary geographical information). The resulting population grids capture both intra-day and monthly population variations at 1 km$^2$, making this dataset the only one of its kind at continental scale. As the method models individual population groups instead of total population counts, it is also thematically richer than what could be achieved with mobile-phone records alone.

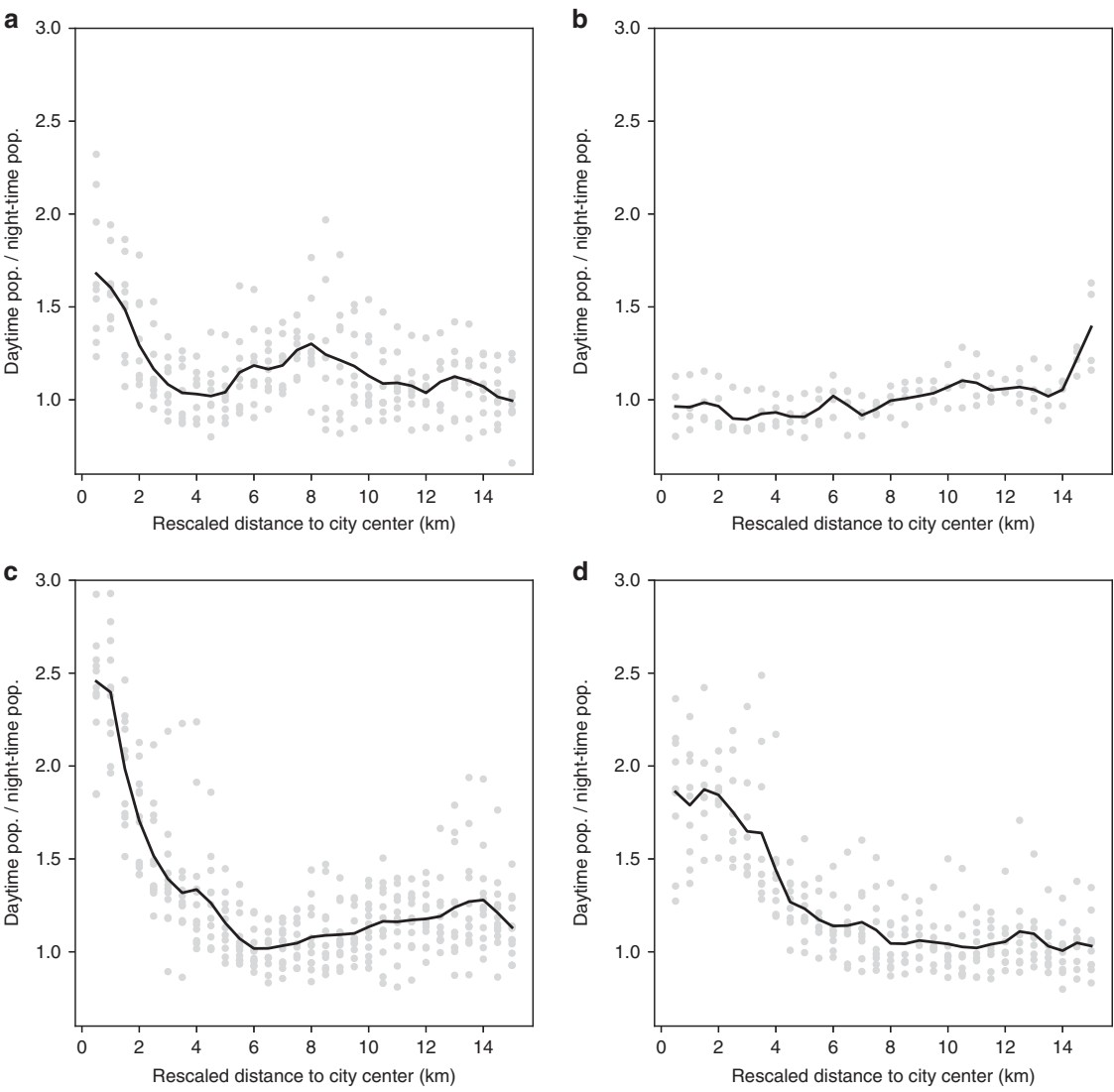

**Fig. 4 Cluster analysis of EU cities.** Ratio between day- and nighttime population densities for the 34 EU cities with a population above 1 million in 2011, grouped in 4 clusters determined using the *k*-means method. Each panel (**a–d**) corresponds to a subset of the 34 cities with a similar spatio-temporal population profile. Source data are provided as a Source Data spreadsheet file.

| Table 1 Summary of the results of the cross-comparison exercise. | | | | | | | | |
|---|---|---|---|---|---|---|---|---|
| **Country** | **Spatial zone type** | **No. of zones** | **Median zone size** | **Reference data source** | **Allocation accuracy** | | **Pearson correlation** | |
| | | | | | **Nighttime** | **Daytime** | **Nighttime** | **Daytime** |
| Italy | Municipalities | 8092 | 21.8 km$^2$ | Census 2011 (ISTAT, Italy) | 99.2% | 92.8% | 1.0 | 0.996 |
| Portugal | Municipalities | 278 | 228.6 km$^2$ | Census 2011 (INE, Portugal) | 99.6% | 92.6% | 1.0 | 0.980 |
| Spain | Municipalities | 53 | 22.1 km$^2$ | Census 2011 (INE, Spain) | 99.3% | 92.3% | 0.999 | 0.999 |
| Belgium | Service areas of mobile-phone towers | 6984 | 3.0 km$^2$ | Mobile Network Operator (Proximus) | 79.8% | 78.0% | 0.866 | 0.849 |

The performed data integration was challenging due to the volume and variety of data in terms of formats, definitions, nomenclatures, and/or semantics. The long and intricate workflow to combine such a variety of data inputs, each with its own inaccuracies, led inevitably to a propagation and accumulation of error in the final product too. Knowing the accuracy of the produced dataset is necessary to inform the users of the product. Therefore, designing a robust quality assessment strategy was no less important and challenging as the modeling per se. The metric

selected for the quality assessment (i.e., allocation accuracy) is a summary metric that compares estimated with reference population for a set of spatial units within a study area. It can be interpreted as the share of the population stock that has been allocated to the correct spatial units. The allocation accuracy is affected by conceptual differences between the two instances being compared. Although our grids represent an ideal or maximum population density at both day- and nighttime, data from mobile-phone operators represent observed mobile-phone user

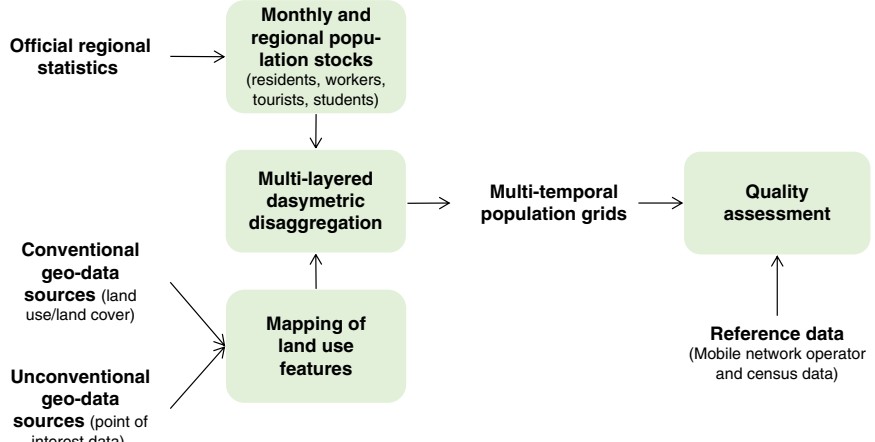

**Fig. 5 Methodological workflow.** Representation of the main classes of input data and methodological steps (green boxes) carried to produce and validate the multi-temporal population grids.

densities from a single mobile network operator in a specific time and day of the year. Therefore, deviations between our estimates and the values from the independent datasets cannot be strictly interpreted as errors. Lacking fully comparable spatio-temporal population records, the cross-comparison exercise cannot be considered a definitive validation, but it corroborated the plausibility of our population grids.

Although these grids are important to enable harmonized and more detailed analyses across national borders in the European context, the herein proposed approach can be transferred to any other region where sufficient data exists (or can be derived) on population stocks and location of activities. Regarding the latter aspect, detailed information contained in global products such as the OpenStreetMap can be of help. The approach is also scalable to the amount of ancillary data available, obviously requiring more adaptation and concessions in data-poor environments.

The novel population grids were originally produced at $100 \times 100$ m cell size but were resampled to $1 \times 1$ km cells for public dissemination. This choice reflects our preference for lower uncertainty of estimates over higher spatial detail and also helps avoid false precision that could potentially lead to overconfidence in the accuracy. In the context of risk mapping and assessment, this cell size is similar or smaller than that of many available hazard data covering the study area (e.g., pan-European seismic map[65]) and is suitable for baseline risk assessment of areas ranging from large urban areas to a whole continent. Notwithstanding, for operational disaster risk management (e.g., preparedness and response) or for hazards with very high local variance, a higher spatial detail would be desirable. In sum, the suitability of the 1 km² resolution equates more to the scale of analysis and desired level of precision than to the domain of application.

Another apparent limitation concerns the reference year of the population data (i.e., 2011). Although certain applications require more up-to-date information, our grids establish a milestone by providing a point of reference for future comparison, namely after the 2021 European censuses become available. Besides, although cities may grow or decline in absolute numbers, their internal spatio-temporal structure is not likely to suffer dramatic changes in short time spans. Tourism seasonality is also rather stable over the years[66]. Therefore, for quick assessments requiring updated population volumes, a rescaling of the herein grids could be an acceptable compromise until new grids reflecting the censuses 2021 are produced.

The work towards spatio-temporal population mapping is just commencing and the potential insights that can be obtained from

such data have just surfaced in this study. It is likely to be that the diversity, quality, and quantity of suitable input data will continue growing[67,68] in tandem with the need for better assessments. For example, certain applications would benefit from population grids stratified by demographic and socioeconomic attributes such as age or income levels, to be effective. To add more dimensions to spatio-temporal population grids, it may be worth investigating whether and how to borrow concepts and methods applied to generate the synthetic populations that underpin agent-based models[69]. Another avenue for future work is the increase of the temporal resolution by integrating temporal signatures of different LU types and activities derived from mobile-phone records.

## Methods

**General framework**. We have developed a multi-layered dasymetric approach that models the spatial distribution of different population groups separately and according to a selection of covariates derived from novel geospatial data sources. The methodology follows four interlinked phases (Fig. 5) as follows: (1) estimation of monthly and regional stocks of population groups; (2) mapping of LU features relevant to the location of the population groups; (3) dasymetric disaggregation of population group stocks to their most likely locations within regions; and (4) quality assessment by means of a cross-comparison against independent datasets for selected countries.

**Estimation of monthly and regional population stocks**. Based on their expected differences in spatial behavior, we distinguish 16 population groups. The basic idea is that a person's location is determined by his or her main activity, as he or she is expected to spend most time there. Students, e.g., can be associated with education facilities (e.g., schools and university campuses) and workers with a range of service and production facilities depending on the economic sector they work in. In practice, we constructed monthly matrices with stocks of these population groups for each NUTS3 region in the study area for the reporting year 2011 (12 matrices, each with dimension = 1311 regions × 16 population groups). The NUTS classification is a hierarchical system of nested territorial units used for statistical data reporting in Europe. The NUTS3 level corresponds to country provinces or districts and comprises 1311 regions within the area of interest, with a median size of 1717 km² (NUTS3 version 2010). NUTS3 are aggregations of municipalities, whereas census zones are small-area units within each municipality. In this study, census zones were only used directly in the cross-comparison exercise explained further down.

The 16 population groups include residents, employees, students, the non-working and non-studying population, and tourists, as detailed next.

Residents correspond to the number of registered residents within a region (obtained from Eurostat at NUTS3 level). Employees are subdivided in 11 economic sectors based on the NACE rev.2 classification of economic activities (Supplementary Table 4) and were obtained from Eurostat reflecting the NUTS3 region of work.

Students are broken down in two main educational levels: primary plus secondary education and tertiary education and above. Student statistics were available from Eurostat at NUTS2 level. Students below tertiary education were distributed among the respective NUTS3 regions based on the proportion of the relevant population age groups. Higher education students were downscaled to

NUTS3 regions based on the number of enrolled students per NUTS3 available from the European Tertiary Education Register, reporting year 2011 (https://www.eter-project.com). In months with more than 50% of school/academic holidays, students were considered part of the non-working and non-studying population group. Country-specific holiday calendars for schools and universities were obtained from existing European-wide inventories[70,71].

The non-working and non-studying (N) population group can be associated with residential areas in both day- and nighttime. It was calculated at NUTS2 level as:

$$N = U + (R - A - S) \qquad (1)$$

based on the number of unemployed, $U$, residents, $R$, active population, $A$, and students, $S$, from Eurostat. $N$ was then downscaled to NUTS3 level proportionally to the population size.

In our approach, the monthly variation of the total present population in a region is primarily linked to inbound and outbound flows of people that visit and leave regions for any purpose, leisure, and business alike. We refer to them as tourists and the estimation of their monthly and regional totals involved several steps. First, annual number of nights spent within each NUTS2 region (Eurostat) were disaggregated to NUTS3 regions proportionally to the number of bed places in touristic accommodations available per NUTS3 from Eurostat. The resulting NUTS3 annual number of nights spent were broken down per month using regional (NUTS2 or NUTS3) seasonal curves constructed from data procured from National Statistical Institutes. Finally, we divided the regional and monthly nights spent by the number of days of each month to obtain the average daily number of inbound tourists. A detailed account of the methodology and input data has been published elsewhere[66].

From National Statistical Institutes and the Organisation for Economic Co-operation and Development, we obtained the share of inbound tourists per country of origin or groups of countries of origin. In the latter case, we split tourists per country of origin employing a model based on geographical distance and economic size (i.e., Gross Domestic Product), assuming larger and closer economies draw higher quantities of tourists. Then we summarized outbound tourists per country of origin. Finally, from Eurostat we obtained the fraction of tourism going outside the EU and added it to the previous sum to obtain the total amount of outbound tourists per EU country. Tourists from countries outside the study area represent added population to the existing stock and therefore did not need any further treatment. Tourists from the same country (domestic) or from countries within the study area (non-domestic) had to be subtracted from their countries and regions of origin to avoid double counting of total population within the study area. The share of outbound tourists per region within each country was assumed to be proportional to their demographic size. The resulting outbound tourists per region were finally subtracted from the different population groups proportional to their size.

**Mapping of land-use features.** LU and LC features are widely used as covariates in dasymetric population mapping[2–4,6,7,20]. In this phase, we constructed the set of geospatial layers to be used as ancillary information in the population dis-aggregation process. Ultimately, we created two distinct types of input data as follows: (a) a fine-grained LULC map and (b) a set of activity density layers.

The LULC map was produced by integrating geospatial data from a wealth of sources. The map is originally based on the CORINE Land Cover (CLC) 2012 map and nomenclature, but achieves a significantly higher thematic and spatial detail. The 11 artificial LU classes from CLC are subdivided in 18 more specific classes, including Production facilities, Commercial or service facilities, Public facilities, and Airport terminals, which were instrumental for the allocation of certain population groups. The minimum mapping unit in this new map is 1 ha for artificial surfaces and 5 ha for others, as opposed to 25 ha in the original CLC data (see Supplementary Note 1 for more details). The production and validation of this novel map has been documented in a dedicated article[72].

Recent findings indicate that the quality of dasymetric population mapping can be increased through the integration of Point of Interest (POI) data[73]. Therefore, complementary to the LULC map, we built a set of activity layers based on POI and polygon data extracted from TomTom Multinet and OpenStreetMap, to represent locations of activities and facilities associated with the presence of students and workers. The selection of these features was based on correspondence with the considered population groups (students and workers from 11 economic sectors). For each population group (e.g., workers in the manufacturing sector), the relevant features (e.g., factories) were processed into a single binary raster layer with a 100 × 100 m resolution and were treated as an additional LU class in the subsequent disaggregation step. For the disaggregation of tourists, we built a layer reporting touristic accommodation room density based on data from online booking platforms[66]. These activity layers were necessary to capture overlapping activities and because conceptual differences between point-based data and the polygon-based LULC map that make their integration difficult. In the Supplementary Note 1, we further discuss the adequacy of the used POI data sources.

**Dasymetric disaggregation of population.** We downscaled each of the 16 population groups originally from NUTS3 regions to 100 m pixels for each month of the year using a two-tier approach. In essence, the stock of a population group

within a region is first divided over the LU types relevant to that group proportional to the occurrence of these LU types within the region (Eq. 2). Consequently, population groups can be associated with multiple LU types (see Supplementary Tables 2 and 3). Conversely, specific LU types may be associated with multiple population groups, in which case they will co-occur in the same LU. In the second step, the number of persons per LU type are allocated to individual grid cells based on built-up density (Eq. 3).

$$P'_{j,r,u} = P_{j,r} * \left( \frac{Q_{r,u} * w_u^j}{\sum_u Q_{r,u} * w_u^j} \right) \qquad (2)$$

$$P'_{j,r,i} = \sum_u \left[ P'_{j,r,u} * \left( \frac{d_{r,u,i}}{\sum_i d_{r,u,i}} \right) \right] \qquad (3)$$

where $P'$ is the estimated population of a given $j$ population group, in pixel $i$ within a NUTS3 region $r$. $Q$ is the count of 100 m grid cells in a region of a given LU class $u$ and $w$ is a Boolean parameter that establishes the link between LU classes and population groups (1 if population group $j$ is linked with LU class $u$, 0 otherwise). The links were based on expert judgment and can be consulted in Supplementary Tables 2 and 3. Finally, $d$ is the built-up density based on the European Settlement Map 2012–release 2017 (https://land.copernicus.eu/pan-european/GHSL/european-settlement-map). In this dataset, built-up density is the percentage of surface covered by all roofed constructions without considering building volumes or density of activities. See Supplementary Note 1 for more details concerning this dataset.

There were some exceptions to this general approach. Residents were downscaled from the 1 km² GEOSTAT grid (https://ec.europa.eu/eurostat/web/gisco/geodata/reference-data/population-distribution-demography/geostat), consistent with the Census 2011, and employing the same rationale as in Eqs. 2 and 3. The non-working and non-studying population layers were obtained by applying NUTS3-specific ratios between the non-working and non-studying, and the number of residents to the number of residents at 100 m level. The total number of tourists per NUTS3 was downscaled twice, generating two distinct grids for each month of the year as follows: (a) one grid reflecting their nighttime distribution (based on the touristic accommodation room density layer mentioned above) and (b) one grid reflecting their daytime distribution (based on a set of LU classes).

In total, the downscaling procedure generates 204 intermediate population grids, i.e., 12 months × 17 population groups (15 population groups + 2× tourists), at a spatial resolution of 100 × 100 m. For each month of the year, the respective nighttime population grid was the result of the sum of the gridded residents with the gridded tourists at nighttime. Conversely, the daytime population grid was the result of the sum of the 15 remainder population group grids (see Eq. 4). The final 24 grids were obtained by aggregating the 100 m pixel values to the target 1 km² grid cells.

$$P'_{i,r,t} = \sum_{j=1}^{n} P'_{j,r,i} \quad \text{where } n(t) = \begin{cases} n = 2 \text{ if } t = \text{'nighttime'} \\ n = 15 \text{ if } t = \text{'daytime'} \end{cases} \qquad (4)$$

Although the final maps are provided at 1 km² resolution, the disaggregation was executed at the native 100 × 100 m resolution of the LULC map to leverage the maximum possible available detail of the input data (resampling the LULC map to 1 × 1 km would result in a gross generalization of the LULC classes). Moreover, this allowed us to preserve the highest possible resolution for more detailed inspection of the results.

**Cross-comparison.** To assess the quality of the produced grids, a series of cross-comparison analyses was performed for areas where independent night- and daytime population estimates at sub-NUTS3 level were available. The comparison was operated at the level of the native spatial units of each independent dataset, here denoted as $m$. For this purpose, our population estimates were aggregated from grid level to the relevant spatial units. For each country $c$ and temporal frame $t$, we computed an accuracy metric herein called allocation accuracy, $AA$ (Eq. 5). It can be interpreted as the percentage of the population stock that has been allocated to the correct spatial units. Metrics based on the sum of absolute errors are common in dasymetric mapping evaluation, because they are more robust in the presence of outliers or for skewed distributions[6,7,74]. In Eq. 5, the asterisk denotes the independent dataset.

$$AA_c^t = \left[ 1 - \left( \frac{\sum_{m=1}^{n} |P' - P_m^{*t}|}{2} \bigg/ \sum_{m=1}^{n} P_m^{*t} \right) \right] * 100 \qquad (5)$$

For Italy, Portugal, and Spain, we obtained Origin–Destination matrices with commuting of students and workers between municipalities from the census carried in 2011. This allowed us to recreate the likely size of the daytime population of each municipality by simply subtracting and adding the number of incoming and outgoing students and workers to the number of residents. These census-based daytime population values do not include tourists. Hence, for the purpose of the cross-comparison, we generated grids that did not take into account inbound and outbound tourists.

We further compared the population totals in our grids for Belgium with a dataset based on cellphone records from the Proximus, the leading mobile network

operator in the country, accounting for nearly 40% of the mobile subscriptions. The number of cellphone records was calculated by Proximus based on signaling data, so capturing connections between the mobile devices and cell towers at high temporal frequency. The dataset consisted of mobile-phone user counts at the level of Voronoi polygons around each cell tower in the country, with a temporal breakdown of 15 min for a specific weekday outside the holiday season (i.e., Thursday, 08/10/2015). Based on this, we produced day- and nighttime frames based on the average observed counts in the periods 9:30–11:30 a.m. and 3:00–5:00 a.m., to capture core working and sleeping hours, respectively. We dissolved Voronoi polygons smaller than 1 sq. km with surrounding polygons to avoid spatial units smaller than the size of our final grids. Although this dataset cannot be used directly to predict total population, as Proximus covers a limited market share in Belgium, we assume it reflects the relative presence of population in space and time. Hence, before applying Eq. 5, we rescaled the number of mobile-phone users to match the total population in Belgium, assuming a constant market share across regions, as in previous studies[47].

**Reporting summary**. Further information on research design is available in the Nature Research Reporting Summary linked to this article.

## Data availability

The multi-temporal population grids for the European Union at 1 km$^2$ resolution that have been generated during this study[75] have been deposited in the European Commission's Joint Research Centre Data Catalog, with identifier 10.2905/BE02937C-5A08-4732-A24A-03E0A48BDCDA, and can be accessed at https://data.jrc.ec.europa.eu/dataset/be02937c-5a08-4732-a24a-03e0a48bdcda. These multi-temporal grids are the source data for Figs. 1 and 2. Source Data are provided with this paper.

## Code availability

The geospatial data processing and analysis were programmed using Matlab and Python, and using standard packages. The results can be reproduced by employing the equations, explanation, and parameters provided in the main text and in the Supplementary Information. Even so, any code produced can be made available upon reasonable request to the authors.

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

## Acknowledgements

The work documented in this study was developed in the context of the "ENhancing ACTivity and population mapping" (ENACT) research project of the European Commission, Joint Research Centre. The European Commission accepts no responsibility or liability whatsoever for the use which may be made of this dataset. The views expressed are purely those of the authors and may not, in any circumstances, be regarded as stating an official position of the European Commission. We thank the Belgium Mobile Network Operator Proximus and Eurostat for making available aggregated spatio-temporal counts of mobile-phone users for the purpose of the cross-comparison.

## Author contributions

F.B.S., S.F., M.C., and C.L. designed the research. F.B.S., S.F., M.S., K.R., and M.A.M.H. performed data collection and preprocessing. M.S. and K.R. performed data fusion. F.B.S., S.F., M.S., and L.Z. performed the model validation. F.B.S. and E.K. analyzed the results. F.B.S. wrote the paper with contributions from all authors.

## Competing interests

The authors declare no competing interests.
