## [Peer Review File · Nature Communications]

Reviewers' comments:

Reviewer #1 (Remarks to the Author):

This is an interesting paper where the authors describe a data fusion based approach to create nighttime residential and daytime population dynamics data for the European Union at 1 km² resolution. This is important because, as the authors mention, there is no single, measurable statistical concept instrumental for daytime population as the number of residents is for night-time population. The authors have provided an adequate background motivation and done a very comprehensive compilation of previous literature that have evidenced the progress of refining official census estimates in space and time. They explain the rationale behind continued attempts by research communities to develop spatiotemporal representation of population with disparate data fusion methods given the access and usage constraints around mobile phone data from the private sector. While the resulting data set is novel for the EU and useful for a wide range of user community, this paper falls short of the scientific and analytical novelty expected for the audience of this journal. Overall, this paper could be focused on one of two topics: (1) development of the data sets including validation and comparison with other available community data for comparison or (2) analysis of EU cities to highlight similarity or lack thereof in urban activity and land use composition. By attempting to do both, it fails to convey a coherent narrative of the novelty.

Title: the title of the paper is not representative of the content and should provide the geographic context i.e. the EU.

Abstract: "Current knowledge of the spatial distribution of population is primarily based upon place of residence statistics, thus ignoring..." Usage of current knowledge is not appropriate. Many similar data sets exist from local to national scales. What I believe the authors want to highlight that official enumeration and collection of census data are still dependent on nighttime residential population.

Method: the authors employ a multilayered dasymetric modeling approach (originally referred to as smart interpolation) with a number of demographic and activity data sets across the EU. The chosen method is well established in the research community and, as the authors point out, has been demonstrated successfully for developing similar data set for the USA in mid-2000s. Given the title of the paper, the approach does not reveal if and how multilayered dasymetric modeling has been extended as a spatial interpolation tool. That will be a key contribution for the community.

The authors skim over a few methodological approaches that are important for explaining the results. For example, on p. 7, "Population densities and distance to center were rescaled as prescribed by Lemoy and Caruso⁵⁷ to remove the effect of city size from the shape of the curves." It is important to mention the effect to explain why that needs to be removed.

Results and Discussions: The authors have done due diligence in analyzing the resultant data for major EU cities. The emergence of new knowledge from the analysis is less than convincing. The fact that daytime population of cities expand up to 1.9 times as compared to nighttime is a quantified assessment of the longstanding qualitative knowledge. Their results for nighttime are consistent with other literature (again, no new knowledge).

Intercomparison and similarity assessment of cities with the k-means is quite interesting. However, a simple Google scholar search will reveal, since early 2000s a number of researchers have argued over the choice of K-means algorithms, particularly for spatial data sets, for providing optimal results and understanding. For example, Density-Based Spatial Clustering of Applications with Noise (DBSCAN), KHM (K Harmonic Means, etc. It is not that k-means is an invalid choice, but authors should provide a clear justification for why k-means was chosen over other methods.

Page 8: what is important to know is not that Lyon, France clusters with three larger Spanish cities, but why? The population data helps to identify that similarity but how could we explain and use that finding?

On page 10, "While our ratios appear lower than what census data suggest (likely due to an underestimation of daytime population density within the Spanish cities), the peculiarity of the

Spanish cities is corroborated, owing to very high residential density of their cores. Most of the discussions are along the lines of explaining the spatial variability of the population distribution with existing land use knowledge. But isn't that expected given land use/activity data were used to develop the disaggregated layers?

This is where I feel the authors miss an opportunity to distinguish their contribution beyond describing development of a new data set for EU.

Availability of high resolution daytime population estimates clearly increases the fidelity of many socioeconomic and engineering applications. However, majority of those applications require demographic attributes along with the population numbers in order to be effective. For example, siting of infrastructures (K-12 schools or nursing homes for the elderly or employment accessibility for working age population) or exposure assessment in public health (breast cancer in women, childhood mortality, etc.). Authors completely miss the opportunity to raise this important limitation in current daytime dataset including their own. Thoughts on how accessibility to such well-developed activity and official census data set can enable creation of daytime demographic attributes is absolutely necessary to mention.

Authors neglect an important reference that first demonstrated a robust approach for bottom up population distribution estimation given the lack of reliable census data.

Weber, E., Seaman, V., Stewart, R., Bird, T., Tatem, A., McKee, J., Bhaduri, B., Moehl, J. and Reith, A., 2018. Census-Independent Population Mapping in Northern Nigeria, *Remote Sensing of Environment*, 204 (786-798), <https://doi.org/10.1016/j.rse.2017.09.024>

Reviewer #2 (Remarks to the Author):

This is a well-designed and well-written study that I think would be of interest to the larger research community. I also believe this research will push the conversation forward for innovative and novel approaches/techniques for producing gridded population products. The authors are straight forward in their assessment, rationale, and breakdown of their work and I commend their effort for taking on the challenge in leveraging different types of population and population proxy data to create products with a finer temporal signature than other gridded products. With that said, I think there is a lot of details that must be addressed before this work can be published. These issues relate to terminology, underlying theory, and methods.

General Points

- In the abstract, need to better drive home imp't of why differences in where people are when matters...perhaps emphasizing (if not in abstract than early on in intro) why this approach needed when temporal granularity finer with other types of data (e.g. CDR). Authors touch on this but I think the authors need to better situate the novelty of this data relative to other types especially considering time stamp of data products.
- Situate in discussion how/why this approach would and would not work in a different context outside of Europe. And if not, why important for the European context (speaking back to application of data).
- In the introduction, unpack and explain how dasymmetric mapping actually lessens issue of MAUP
- Also in the introduction, reference specific to citation 5, Leyk et al review, for more detailed ref on differences between top down gridded population data sets.
- Reliability of produced grids gets organized into the discussion instead of methods – not sure that makes the best sense in terms of organization. As a reader, it felt like I was back tracking in the reading.

Terminology

- I appreciate the sentiment that data integration was challenging and kudos to the authors for taking on the task of working across a wide spread of data formats. Agree that definitions can be debated...perhaps include justification for choice of city definition from Eurostat. What is this option

the best choice?

- Why the use of the term stock? Other studies reference source and target areas to denote origin spatial grain and gridded output...
- More detail regarding the term built-up density and the layer associated with its' representation. Clicking the provided link notes that the product is produced with SPOT 5 and 6 multi-spectral satellite products combined with OSM data. Correct? I think interested readers will want additional detail here regarding the built up density product since population is then allocated to those pixels. Also, generally speaking, I was under the impression the term "built-up" represented a raster settlement layer informed somehow with vertical information/3-d building structures, whether radar or some other data.

Theory

- "A common way to characterize urban densities is by creating density gradients that describe the decay of population densities as a function of distance to city centers⁵⁶" - How does this intersect with considerations that characterizing urban areas (i.e. built/built up) may have much more fragmented or dispersed patterns, different cities have different sprawl patterns, intensification may be more prevalent in one context versus another...in which case, a one type fits all function of distance from city center would be a poor way to characterize associated population pattern...I think the authors need to better support using decay as a fxn of city center distance for characterizing population densities in an urban environment

Methods

- "Finally, night- and daytime-specific subsets of intermediate grids were summed to obtain the final monthly day- and night-time population grids" Meaning, matched back to totals from aggregate level, or stocks of the individual pop groups?
- Usefulness of 1 km² for final population gridded maps. What does adequate really mean? For climate studies this spatial grain is high but for many other applications that involve disaster related, hazard-related, health-related events/risks/etc, 1 km² seems pretty coarse...Disconnect to above application rationale and rationale provided here for spatial grain (i.e. regional analyses and urban use), need to better relate grid cell size of products to end use
- Population census data description and context on how tied to the NUTS classification of spatial units needs to be better detailed. Also, explanation to help bridge the disconnect between using 2011 as a reference year and production year for the maps in order to be consistent with the last round of the European censuses and how maps showing differences of night/day population from almost a decade ago help with disaster risk management, planning transport and social infrastructure...is the assumption that population totals haven't changed that much? Movement across the EU is comparable in 2020 to 2011? How easy would it be for a researcher/end user of this data to extrapolate findings to more contemporary date?
- Need more detail on the multi-fine scale LULC map produced for the modeling. While nitty gritty details in the production and validation are fine in another article, there needs to be more detail in current manuscript on how the 18 specific classes were determined and why difference in spatial grain of artificial and non-artificial surfaces, how those surfaces assume to relate to various land uses, etc.
- Quality of the TomTom and OSM data used for given areas? Vetted by other sources? Assumptions inherent in use? Temporal relation with 2011 census data?
- "We downscaled each of the 16 population groups originally from NUTS3 regions to 100 m pixels for each month of the year using the two-tier approach" – why if final maps at 1 km?
- How do you deal with land use types that overlap population groups? Proportional to "their" occurrence in the region – meaning the occurrence of the land use or the population group?
- "temporal breakdown of 15 minutes for a specific week-day (08/10/2015)" – significance of this weekday chosen along with times chosen for day and night time movement? How limited a market share does Proximus cover?
- Figure 1 should give a sense of total counts tied to bars. It would be nice to situate Milan in larger background map for location. Or a scale bar. Both would be preferable.

Review of the manuscript “The changing population densities in time and space: a data fusion approach for Europe”, submitted to Nature Communications (NCOMMS-20-01165)

Reviewer 1 comments

“This is an interesting paper where the authors describe a data fusion based approach to create nighttime residential and daytime population dynamics data for the European Union at 1 km² resolution. This is important because, as the authors mention, there is no single, measurable statistical concept instrumental for daytime population as the number of residents is for night-time population. The authors have provided an adequate background motivation and done a very comprehensive compilation of previous literature that have evidenced the progress of refining official census estimates in space and time. They explain the rationale behind continued attempts by research communities to develop spatiotemporal representation of population with disparate data fusion methods given the access and usage constraints around mobile phone data from the private sector.”

Issue no.	Comment	Author’s response
1.1	While the resulting data set is novel for the EU and useful for a wide range of user community, this paper falls short of the scientific and analytical novelty expected for the audience of this journal. Overall, this paper could be focused on one of two topics: (1) development of the data sets including validation and comparison with other available community data for comparison or (2) analysis of EU cities to highlight similarity or lack thereof in urban activity and land use composition. By attempting to do both, it fails to convey a coherent narrative of the novelty.	We agree with the point raised by the reviewer and have adapted the manuscript to further emphasize the main contribution of this paper which is – as the reviewer points out – the development of the novel dataset. We have downplayed the analysis of the EU cities but not removed it. We consider it is important to keep this topic in the paper but essentially as illustration of the potential and validity of the new dataset. In fact, the confirmation and quantification of knowledge concerning the spatio-temporal structure of cities (as the reviewer points out in issue no. 1.6) helps corroborate the validity of the dataset. To stress the novelty and re-position the paper around the development of the dataset, we modified the abstract, introduction and discussion as necessary. For example, in the introduction we have made additions to better position our contribution vis-a-vis other

		spatiotemporal data sources such as mobile phone records (also per the request of Reviewer 2). We have emphasized that our dataset is the only one that captures both intraday and seasonal population variations at continental scale, and that we achieved that by using novel elaboration of the dasymetric methods, which we refer to as ‘multi-layered dasymetric mapping’. Finally, to further weigh the aspects inherent to the novel dataset, we moved the quality assessment to the results section.
1.2	Title: the title of the paper is not representative of the content and should provide the geographic context i.e. the EU.	We followed the suggestion of the reviewer and changed the title to “The changing population densities in time and space: a data fusion approach for Europe”. Please note that we added ‘for Europe’ and removed word ‘urban’ (to keep the word-count low, but also because the population grids cover the whole territory, and not just ‘urban areas’). In the first version, the word ‘urban’ had been inserted in linkage to our analysis of spatio-temporal structure of cities and which we downplayed (see point 1.1).
1.3	Abstract: “Current knowledge of the spatial distribution of population is primarily based upon place of residence statistics, thus ignoring...” Usage of current knowledge is not appropriate. Many similar data sets exist from local to national scales. What I believe the authors want to highlight that official enumeration and collection of census data are still dependent on nighttime residential population.	We have re-written the first sentence of the abstract: “The knowledge of the spatial distribution of population is often based upon place-of-residence statistics from official sources (e.g. censuses), thus ignoring (...)”.
1.4	The authors employ a multilayered dasymetric modeling approach (originally referred to as smart interpolation) with a number of demographic and activity data sets across the EU. The chosen method is well established in the research community and, as the authors point	We have inserted a reference to ‘smart areal interpolation’ in the 2nd paragraph of the introduction. We have made it more explicit that our ‘multi-layered’ dasymetric approach is in fact an extension of the most common applications of

	out, has been demonstrated successfully for developing similar data set for the USA in mid-2000s. Given the title of the paper, the approach does not reveal if and how multilayered dasymetric modeling has been extended as a spatial interpolation tool. That will be a key contribution for the community.	the dasymetric mapping methods. We have not assembled the algorithm as an open and ready-to-use tool/software – if this is what the reviewer was wondering. One of the reasons for this was that our code was very tailored to the very specific input data available for Europe, and would require adaptation in other geographical contexts. We agree that the release of a stable and functional piece of software would be a plus to the community, but that is beyond the scope of this paper. Moreover, we believe that our paper will draw further attention to the dasymetric modelling and will hopefully encourage others to replicate and/or generalize our approach. As per the code availability section, our code can be shared upon any reasonable request.
1.5	The authors skim over a few methodological approaches that are important for explaining the results. For example, on p. 7, “Population densities and distance to center were rescaled as prescribed by Lemoy and Caruso to remove the effect of city size from the shape of the curves.” It is important to mention the effect to explain why that needs to be removed.	We agree that a simple reference to the work of Lemoy and Caruro (2018) (who dedicated a whole paper to the issue at stake) is insufficient. A lengthier justification to the rescaling of our data is needed to help the reader interpret the obtained results. So, we have added the following extra lines to the relevant paragraph in the results section: “Population densities and distance to center were rescaled as prescribed by Lemoy and Caruso⁵⁹ to make the radial population density profiles comparable across cities of different population sizes, which, in our sample, range over one order of magnitude. The rescaling makes all cities comparable in dimension to the most populous city in the sample (i.e. Paris). Without rescaling, the curves cannot be compared across x and y axes, as more populous cities tend to be denser as well as extend over a larger geographical area to accommodate the extra population⁵⁹.”

1.6	The authors have done due diligence in analyzing the resultant data for major EU cities. The emergence of new knowledge from the analysis is less than convincing. The fact that daytime population of cities expand up to 1.9 times as compared to nighttime is a quantified assessment of the longstanding qualitative knowledge. Their results for nighttime are consistent with other literature (again, no new knowledge).	As explained in issue no. 1.1, we have repositioned the analysis of spatio-temporal structure of EU cities as an illustration of the resulting dataset, and do not claim for the novelty of our insights. However, we do think our paper offers a systematic and unprecedented quantification for Europe of earlier qualitative knowledge. To the best of our knowledge, it is the first time a quantification of the day- to night-time population ratio is done in a complete and comparable manner to the 30+ largest EU cities. We dare say this quantification of the otherwise very qualitative knowledge merits publishing.
1.7	Intercomparison and similarity assessment of cities with the k-means is quite interesting. However, a simple Google scholar search will reveal, since early 2000s a number of researchers have argued over the choice of K-means algorithms, particularly for spatial data sets, for providing optimal results and understanding. For example, Density-Based Spatial Clustering of Applications with Noise (DBSCAN), KHM (K Harmonic Means, etc. It is not that k-means is an invalid choice, but authors should provide a clear justification for why k-means was chosen over other methods.	K-means is a general-purpose, non-supervised classifier adequate for situations with not too many clusters, which is clearly our case, since we want to classify a sample of only 34 cases (i.e. cities). K-means requires the number of clusters as the only parameter, and offers ways to determine the most adequate number of clusters, which is also convenient in our case. Also note that K-means was not run on a pure spatial dataset. Although the dataset was constructed based on spatial data, it suffered various transformations, making it somewhat aspatial: reduced dimensionality, from 2D (coordinates) to 1D (simply distance to city centre); and rescaling of distance to city centre. We have updated the relevant paragraph (8th) of the results section to integrate these considerations. The classification we obtained using K-means, and which is documented in the manuscript, shows marked differences between clusters while modest variance within each cluster, which in our opinion fits the purpose of the classification.
1.8	Page 8: what is important to know is not that Lyon, France clusters	Following your suggestion to stress the development of our new

	with three larger Spanish cities, but why? The population data helps to identify that similarity but how could we explain and use that finding?; On page 10, “While our ratios appear lower than what census data suggest (likely due to an underestimation of daytime population density within the Spanish cities), the peculiarity of the Spanish cities is corroborated, owing to very high residential density of their cores”. Most of the discussions are along the lines of explaining the spatial variability of the population distribution with existing land use knowledge. But isn’t that expected given land use/activity data were used to develop the disaggregated layers? This is where I feel the authors miss an opportunity to distinguish their contribution beyond describing development of a new data set for EU.	dataset, we have limited the attention to the analysis part and taken out the indeed incomplete discussion of possible reasons for the observed phenomena/patterns. We may come back to that in another paper.
1.9	Availability of high resolution daytime population estimates clearly increases the fidelity of many socioeconomic and engineering applications. However, majority of those applications require demographic attributes along with the population numbers in order to be effective. For example, siting of infrastructures (K-12 schools or nursing homes for the elderly or employment accessibility for working age population) or exposure assessment in public health (breast cancer in women, childhood mortality, etc.). Authors completely miss the opportunity to raise this important limitation in current daytime dataset including their own. Thoughts on how accessibility to such well-developed activity and official census data set can enable creation of daytime demographic attributes is absolutely necessary to mention.	Thank you for this valid and relevant point. We have substantially revamped the discussion section (also per the request of Reviewer 2), and took the opportunity to refer to this specific limitation of population grids in general and of our own too. Please see the last paragraph of the discussion where we added: “‘It is likely that the diversity, quality and quantity of suitable input data will continue growing in tandem with the need for better assessments. For example, certain applications would benefit from population grids stratified by demographic and socioeconomic attributes such as age or income levels in order to be effective. In order to add more dimensions to spatio-temporal population grids it may be worth investigating whether and how to borrow concepts and methods applied to generate the synthetic populations that underpin agent-based models.’”
1.10	Authors neglect an important reference that first demonstrated a robust	Thank you for the reference to this very interesting and relevant

approach for bottom up population distribution estimation given the lack of reliable census data. Weber, E., Seaman, V., Stewart, R., Bird, T., Tatem, A., McKee, J., Bhaduri, B., Moehl, J. and Reith, A., 2018. Census-Independent Population Mapping in Northern Nigeria, Remote Sensing of Environment, 204 (786-798), https://doi.org/10.1016/j.rse.2017.09.024	work. We have added an explicit consideration plus the respective reference to the 4th paragraph of the introduction which discusses the situation in data-poor countries.
--	--

Reviewer 2 comments

“This is a well-designed and well-written study that I think would be of interest to the larger research community. I also believe this research will push the conversation forward for innovative and novel approaches/techniques for producing gridded population products. The authors are straight forward in their assessment, rationale, and breakdown of their work and I commend their effort for taking on the challenge in leveraging different types of population and population proxy data to create products with a finer temporal signature than other gridded products. With that said, I think there is a lot of details that must be addressed before this work can be published. These issues relate to terminology, underlying theory, and methods.”

Issue no.	Comment	Author's response
2.1	In the abstract, need to better drive home impt of why differences in where people are when matters...perhaps emphasizing (if not in abstract than early on in intro) why this approach needed when temporal granularity finer with other types of data (e.g. CDR). Authors touch on this but I think the authors need to better situate the novelty of this data relative to other types especially considering time stamp of data products.	We agree with the reviewer and have changed both the abstract and the introduction to accommodate these suggestions. In the introduction we compared more thoroughly the pros and cons of CDR (and more generally mobile phone) data, and why our approach is necessary. In short, we argue that, although mobile phone data can potentially provide higher temporal granularity, its use is constrained by very limited data accessibility and biases that compromise quality and comparability. Instead, our approach guarantees seamless and harmonized continental coverage, at low cost. In the revised discussion, we suggest that the combination of our approach with sample data from mobile network operators can be considered in future work, namely to improve the temporal resolution.
2.2	Situate in discussion how/why this approach would and would not work in a different context outside of Europe. And if not, why important for the European context (speaking back to application of data).	We have added this relevant point to the revised discussion section of the manuscript (see 4th paragraph of the discussion): “While these grids are important to enable harmonized and more detailed analyses across national borders in the European context, the herein proposed approach can be transferred to any other region

		where sufficient data exists (or can be derived) on population stocks and location of activities. Regarding the latter aspect, detailed information contained in global products such as the OpenStreetMap can be of help. The approach is also scalable to the amount of ancillary data available, obviously requiring more adaptation and concessions in data-poor environments.”
2.3	In the introduction, unpack and explain how dasymetric mapping actually lessens issue of MAUP.	We have now duly explained how dasymetric mapping can lessen the issue of MAUP when used to generate population grids. The improved text in the 3 rd paragraph of the introduction goes as follows: “Dasymetric mapping is often applied to generate 'population grids', or tessellations of regular squared cells with estimates of population. Such grids help mitigate the distortions associated with the Modifiable Areal Unit Problem ¹⁶ to the extent they increase the spatial resolution, are less arbitrary and remove the original areal heterogeneity vis-à-vis the original population enumeration zones.”
2.4	Also in the introduction, reference specific to citation 5, Leyk et al review, for more detailed ref on differences between top down gridded population data sets.	We inserted the requested reference at the end of the 4 th paragraph of the introduction: “(…) for a recent review more centered on top-down methods for large-scale applications and their fitness for use, we recommend the paper by Leyk and colleagues.”
2.5	Reliability of produced grids gets organized into the discussion instead of methods – not sure that makes the best sense in terms of organization. As a reader, it felt like I was back tracking in the reading.	We agree with the reviewer that the quality assessment was inconveniently placed in the discussion. We have reorganized the manuscript to improve readability. In the revised manuscript, the methods section (last subsection) describes the data and methods used in the quality assessment, while the actual results of the quality assessment got moved from the

		discussion to the results section, in a dedicated subsection. However, we kept in the discussion section the paragraph that addresses the limitations of the performed quality assessment.
2.6	I appreciate the sentiment that data integration was challenging and kudos to the authors for taking on the task of working across a wide spread of data formats. Agree that definitions can be debated...perhaps include justification for choice of city definition from Eurostat. What is this option the best choice?	We provided extra background on the merits of the selected city definition (4 th paragraph of the results section): “(...) we investigated some spatio-temporal characteristics of the largest urban agglomerations in Europe. To select them, we used the city/greater city extents defined by Eurostat ⁵⁷ . This definition was designed to improve comparability of city statistics, and applies a fixed set of criteria related to urban morphology to consistently characterize city limits irrespective of national definitions.”
2.7	Why the use of the term stock? Other studies reference source and target areas to denote origin spatial grain and gridded output...	We do not use the term ‘stock’ in substitution of the term ‘source zones’. The term stock refers to the size of each population group within each source zone. Although we chose not to use the terms ‘source’ and ‘target’ zones, in our application they correspond to the NUTS3 regions and the 1 km ² grid cells, respectively. We believe the manuscript is sufficiently clear to what is the meaning and methodological use of these concepts.
2.8	More detail regarding the term built-up density and the layer associated with its’ representation. Clicking the provided link notes that the product is produced with SPOT 5 and 6 multi-spectral satellite products combined with OSM data. Correct? I think interested readers will want additional detail here regarding the built up density product since population is then allocated to those pixels. Also, generally speaking, I was under the impression the term “built-up” represented a raster settlement layer informed somehow with vertical information/3-	We have provided more explanation to the actual meaning of built-up density as captured by the used dataset (European Settlement Map). In the subsection “Dasymetric disaggregation of population”, it now reads: “Finally, d is the built-up density based on the European Settlement Map 2012 – release 2017. In this dataset, built-up density is the percentage of surface covered by all roofed constructions without considering building volumes or density of activities. See

	d building structures, whether radar or some other data.	Supplementary Text 1 for more details concerning this dataset.” For the interested reader, we added further information and references in the the Supplementary Text 1 concerning specifically the European Settlement Map. We would have gladly used built-up information that captures volume of built-up instead of simple horizontal density. Unfortunately, to the best of our knowledge, no such layer exists or is available yet for the whole of Europe.
2.9	“A common way to characterize urban densities is by creating density gradients that describe the decay of population densities as a function of distance to city centers” - How does this intersect with considerations that characterizing urban areas (i.e. built/built up) may have much more fragmented or dispersed patterns, different cities have different sprawl patterns, intensification may be more prevalent in one context versus another...in which case, a one type fits all function of distance from city center would be a poor way to characterize associated population pattern...I think the authors need to better support using decay as a fxn of city center distance for characterizing population densities in an urban environment.	Our analysis is not meant to comprehensively characterize urban areas and urban morphology (there is a whole body of literature on urban morphology), but it is a essentially an illustration of how well our data is able to confirm known relations. In a sense, it helps validate our dataset too. We agree with the reviewer that the spatial distribution of population density in urban settings will show variation per city based on local conditions, and there are various ways to capture this. However, we note that the relationship between population density and distance to city centers (i.e. the concentric population density gradient) is remarkably stable as noted by the work of Lemoy and Caruso 2018) and confirmed by our data too, as indicated by the high R^2 in Figure 3. In line with the above considerations, we added the following text to the manuscript (results section, paragraph 6): “Although population density distribution will show great variation per city based on local conditions, its relation with the distance to city centers is remarkably stable within our ensemble of cities, as

		indicated by the high R^2 obtained (0.99) (Figure 3).”
2.10	“Finally, night- and daytime-specific subsets of intermediate grids were summed to obtain the final monthly day- and night-time population grids” Meaning, matched back to totals from aggregate level, or stocks of the individual pop groups?	We acknowledge the sentence was not very clear. We have rephrased and provided an example for additional clarity. The sentence in the 1st paragraph of the results sections now reads: “To obtain the final monthly day- and night-time population grids, we summed the respective monthly grids of specific population groups. For example, the daytime population grid for January corresponds to the sum of the previously generated grids for January of workers, students, tourists and the non-working & non-studying population (refer to the Methods section for a more detailed explanation).” Please note that this paragraph in the beginning of the results section is intended to give only a very brief description of the methodology. The details concerning the input data and how these have been combined are provided in the methods section.
2.11	Usefulness of 1 km ² for final population gridded maps. What does adequate really mean? For climate studies this spatial grain is high but for many other applications that involve disaster related, hazard-related, health-related events/risks/etc, 1 km ² seems pretty coarse... Disconnect to above application rationale and rationale provided here for spatial grain (i.e. regional analyses and urban use), need to better relate grid cell size of products to end use.	We agree that this point deserved elaboration in the paper. Since it relates to the potentialities and limitations of the produced dataset, we have chosen to address it in the discussion section (5th paragraph): “The novel population grids were originally produced at 100 m², but they were resampled to 1 km² for public dissemination. This choice reflects our preference for lower uncertainty of estimates over higher spatial detail, and also helps avoid false precision that could potentially lead to overconfidence in the accuracy. In the context of risk mapping and assessment, this cell size is similar or smaller than that of many available hazard data covering the study area (e.g. pan-European seismic map⁶³) and is suitable for baseline risk assessment of areas ranging from large urban areas to a whole continent.

		Notwithstanding, for operational disaster risk management (e.g. preparedness, response), or for hazards with very high local variance, higher spatial detail would be desirable. In sum, the suitability of the 1 km² resolution equates more to the scale of analysis and desired level of precision than to the domain of application.”
2.12	Population census data description and context on how tied to the NUTS classification of spatial units needs to be better detailed. Also, explanation to help bridge the disconnect between using 2011 as a reference year and production year for the maps in order to be consistent with the last round of the European censuses and how maps showing differences of night/day population from almost a decade ago help with disaster risk management, planning transport and social infrastructure...is the assumption that population totals haven't changed that much? Movement across the EU is comparable in 2020 to 2011? How easy would it be for a researcher/end user of this data to extrapolate findings to more contemporary date?	Please note that the censuses data are used indirectly in our disaggregation, taking Eurostat as the main source of population groups per NUTS3 level, or using Eurostat's GEOSTAT 1x1 km grid as source for the disaggregation of the number of residents, and which reflects the results of the European censuses in 2011. For those less familiar, the link between census zones and the NUTS classification has been clarified in the paper when we first refer to the NUTS system (methods section, 2nd paragraph): “The NUTS3 level corresponds to country provinces or districts (...). NUTS3 are aggregations of municipalities, while census zones are small area units within each municipality. In this study, census zones were only used directly in the cross-comparison exercise explained further down.” With regards to the almost one decade-old data, we agree it is a fair point, deserving the owed attention in the revised discussion section (6th paragraph): “Another apparent limitation concerns the reference year of the population data (i.e. 2011). Although certain applications require more up-to-date information, our grids establish a milestone by providing a point of reference for future comparison, namely after the 2021 European censuses become available. Besides, while cities may

		grow or decline in absolute numbers, their internal spatiotemporal structure is not likely to suffer dramatic changes in short time spans. Tourism seasonality is also rather stable over the years ⁶⁴ . Therefore, for quick assessments requiring updated population volumes, a rescaling of the herein grids could be an acceptable compromise until new grids reflecting the censuses 2021 are produced.”
2.13	Need more detail on the multi-fine scale LULC map produced for the modeling. While nitty gritty details in the production and validation are fine in another article, there needs to be more detail in current manuscript on how the 18 specific classes were determined and why difference in spatial grain of artificial and non-artificial surfaces, how those surfaces assume to relate to various land uses, etc.	We have complemented the Supplementary Information (Text 1 – Land use/land cover map) with additional details on the LULC map as requested.
2.14	Quality of the TomTom and OSM data used for given areas? Vetted by other sources? Assumptions inherent in use? Temporal relation with 2011 census data?	We have complemented the Supplementary Information (Text 1 – Points of Interest from OpenStreetMap and TomTom) with additional considerations concerning assumptions, data quality and temporal relation with reference year 2011.
2.15	“We downscaled each of the 16 population groups originally from NUTS3 regions to 100 m pixels for each month of the year using the two-tier approach” – why if final maps at 1 km?	This is in fact a relevant question related to our methodology that was left unexplained in the original version of the manuscript. We added the following text to the methods section (under “Dasymetric disaggregation of population”): “Although the final maps are provided at 1 km ² resolution, the disaggregation was executed at the native 100 m ² resolution of the LULC map to leverage the maximum possible available detail of the input data (resampling the LULC map to 1 x 1 km would result in a gross generalization of the LULC classes). Moreover, this allowed us to preserve the highest possible resolution for more detailed

		inspection of the results.”
2.16	How do you deal with land use types that overlap population groups? Proportional to “their” occurrence in the region – meaning the occurrence of the land use or the population group?	Please refer to our improved and expanded explanation in the methods section (under “Dasymetric disaggregation of population”), where we hopefully make the answer to this question more explicit: “We downscaled each of the 16 population groups originally from NUTS3 regions to 100 m pixels for each month of the year using a two-tier approach. In essence, the stock of a population group within a region is first divided over the land-use types relevant to that group proportional to the occurrence of these land-use types within the region (eq. 2). Consequently, population groups can be associated with multiple land-use types (see Supplementary Tables 2 and 3). Conversely, specific land-use types may be associated with multiple population groups, in which case they will co-occur in the same land use. In the second step, the number of persons per land-use type are allocated to individual grid cells based on built-up density (eq. 3).”
2.17	“temporal breakdown of 15 minutes for a specific week-day (08/10/2015)” – significance of this weekday chosen along with times chosen for day and night time movement? How limited a market share does Proximus cover?	Proximus is the leading mobile network operator (MNO) in Belgium, accounting for nearly 40% of mobile subscriptions. The date of the sample (8/10/2015) was the one kindly made available by the MNO. It corresponds to a Thursday outside holiday season. So we are confident it is a good proxy for usual weekdays. As for the chosen day- and night-time ‘temporal windows’ for the aggregation of MNO data, they represent core working hours and core sleeping hours to match the specification of our population grids (see specifications of our grids in the 2nd paragraph of the results section, where we state “The night-time frames represent an ‘ideal’ situation assuming the whole population is at their place of residence

		or lodging to rest, whereas the daytime frames represent a situation whereby everybody is assumed to be at the location of their primary activity such as working or studying during core working hours. As such, in-between daily variations of population are not taken into account (e.g. commuting, pre- or after-work activities, etc.).”). The text was updated accordingly in the methods section (under the cross-comparison subsection).
2.18	Figure 1 should give a sense of total counts tied to bars. It would be nice to situate Milan in larger background map for location. Or a scale bar. Both would be preferable.	Thank you for this useful suggestion to improve the interpretation of Figure 1. As requested, we have added a small context map to the upper right corner of the figure. In addition, we have enriched the caption with information concerning the size of the represented area (150 km x 100 km) and the population volume of the bars. With regards to the spatial scale, please note that each bar corresponds to a 1 km x 1 km thus giving the best sense of scale. This information is also provided in the caption of the figure.

REVIEWERS' COMMENTS:

Reviewer #1: Comments only visible to editor

Reviewer #2 (Remarks to the Author):

I have reviewed the revised submission materials for NCOMMS-20-01165A and I am satisfied with the response and edits made by the authors. The only minor point would be to push the authors to improve Figure 1. It can be better. It's the first figure in the MS and could be greatly improved with some creativity. If nothing else, please give some context to the regional inset (country names, scale bar, etc).